# Using a Battery of Bioassays to Assess the Toxicity of Wastewater Treatment Plant Effluents in Industrial Parks

**DOI:** 10.3390/toxics11080702

**Published:** 2023-08-14

**Authors:** Bin Yang, Haiyan Cui, Jie Gao, Jing Cao, Göran Klobučar, Mei Li

**Affiliations:** 1State Key Laboratory of Pollution Control and Resource Reuse, School of Environment, Nanjing University, Nanjing 210023, China; 2Department of Biology, Faculty of Science, University of Zagreb, Rooseveltov trg 6, 10000 Zagreb, Croatia

**Keywords:** wastewaters, genotoxicity, bioassays, cytotoxicity

## Abstract

Bioassays, as an addition to physico-chemical water quality evaluation, can provide information on the toxic effects of pollutants present in the water. In this study, a broad evaluation of environmental health risks from industrial wastewater along the Yangtze River, China, was conducted using a battery of bioassays. Toxicity tests showed that the wastewater treatment processes were effective at lowering acetylcholinesterase (AChE) inhibition, HepG2 cells’ cytotoxicity, the estrogenic effect in T47D-Kbluc cells, DNA damage of Euglena gracilis and the mutagenicity of Salmonella typhimurium in the analyzed wastewater samples. Polycyclic aromatic hydrocarbons (PAHs) were identified as potential major toxic chemicals of concern in the wastewater samples of W, J and T wastewater treatment plants; thus, the potential harm of PAHs to aquatic organisms has been investigated. Based on the health risk assessment model, the risk index of wastewater from the industrial parks along the Yangtze River was below one, indicating that the PAHs were less harmful to human health through skin contact or respiratory exposure. Overall, the biological toxicity tests used in this study provide a good basis for the health risk assessment of industrial wastewater and a scientific reference for the optimization and operation of the treatment process.

## 1. Introduction

The complexity of water pollution is becoming an emerging concern due to the numerous pollutants that enter water bodies in China. The primary sources of water pollution are untreated industrial and agricultural wastewater, domestic sewage and waste [1]. The wastewater discharged from industrial parks mainly manifests in large volumes, complex compositions and high concentrations of pollutants. The persistent pollutants in wastewater can enter the food chain and ultimately endanger human health [2]. In China, the total national wastewater discharge in 2020 was 71.62 billion tons, of which 29% (20.53 billion tons) was contributed from industrial wastewater discharge. Jiangsu Province in China has 58 chemical industry parks of various industrial scales. Therefore, industrial wastewater serves as an essential source of freshwater and marine pollution [3]. This requires an elaborate environmental risk assessment of industrial wastewater pollution using mandatory biological monitoring as an addition to already existing chemical monitoring.

Bioassays are promising methods for studying these sources of pollution, since, on the one hand, all parameters related to the exposed organisms can be controlled as in laboratory experiments. On the other hand, work under environmentally realistic conditions considers the interactions that occur between the chemicals in the effluent and the complexity of the receiving environment [4]. Recent studies have shown that hematological parameters are often used as valuable indicators for assessing fish health, and that the use of pelagic fish data allows for comprehensive monitoring studies of effluents [5,6]. The effects of wastewater treatment plant effluents on biological neurotoxicity, cytotoxicity, genotoxicity and estrogenic effects have been reported in recent decades [7]. These studies of the impact of the effluents on specific physiological functions provide a complete assessment of the overall health of the organism. Additionally, they provide a more comprehensive perspective by accurately predicting the biological effects of wastewater treatment plant effluents [8]. 

In 2004, the United States Environmental Protection Agency (USEPA) included the Whole Effluent Toxicity Test (WETT) into the implementation guidelines for total wastewater toxicity. The Continental European Organization and Oceania scholars further proposed the Whole Effluent Assessment (WEA) to evaluate the persistence, bioaccumulation and comprehensive toxicity of discharged wastewater using a suite of acute and chronic toxicity tests [9]. Currently, China’s monitoring of toxic substance discharge in industrial wastewater is dominated by the physical and chemical monitoring of pollutants. Such methods and monitoring techniques for chemical industry emissions do not fully represent the possible toxicity of industrial wastewater, which jeopardizes environmental safety and industry development [10]. Therefore, performing and standardizing toxicity testing is one of the critical parts of environmental risk assessment and management in China. 

This study aims to combine chemical and biological analysis of industrial wastewater before and after WWT (influent and effluent samples) along the Yangtze River in Jiangsu Province. The basic physical and chemical properties of the wastewater samples were measured as a basis for the ecological and health risk assessment of wastewater pollution in these industrialized areas. Biological assays were performed at different biological organizations to evaluate the environmental risk of wastewater from chemical industrial parks. Genotoxicity assays were performed using two organisms, freshwater algal species Euglena gracilis and Salmonella typhimurium. Assays were also performed on human hepatocellular carcinoma cell line HepG2 and human breast cancer cell line T47D-Kbluc to determine cytotoxic and estrogenic cellular responses, respectively, to wastewater pollution. Enzyme activity assays were also performed to elucidate the inhibitory response of acetylcholinesterase.

## 2. Materials and Methods

### 2.1. Sample Collection and Preparation

Sampling was performed in accordance with “Water Quality Sampling Technical Guidance” (HJ404–2009). Sampling was conducted using a steel bucket on sunny and rainless days between 10 April and 20 April 2018. Nine inlet and nine outlet samples of the wastewater were collected from the wastewater treatment plants between 9 a.m. and 2 p.m. and samples only contained plant wastewater. Sampling container was sanitized before and after collection to avoid contamination. These WWTPs (W, J and T, as shown in Figure 1) are in the three typical industrial areas along the Yangtze River. A total of nine samples (20 L) were stored under refrigeration at 4 °C for 24 h before carrying out the experiments. The collected samples were processed as follows: filtration (0.7 μm glass fiber filter) was performed to remove particles, followed by acidification at pH 2–3 with concentrated hydrochloric acid. Samples were further processed using solid-phase extraction (Oasis Hydrophile-Lipophile Balance cartridge) and then stored at −20 °C, protected from light, for chemical analysis and bioassays, respectively. Solid-phase extraction was performed in accordance with “Water Quality-Determination of polycyclic aromatic hydrocarbons by liquid-liquid extraction and soild-phase extraction high performance liquid chromatography” (HJ478-2009). The extraction procedure was the following: 10 mL methanol and 10 mL acetone/hexane (*v/v*:1/1) for activating the extraction column (Waters Oasis HLB), 10 mL methanol and 10 mL acetone/hexane (*v/v*:1/1) for eluting, 5 mL/min for solution loading. Before and during the analysis, method blanks, instrumental blanks and solvent blanks were implemented for each batch of samples. Spiked matrices showed 80~110% recovery for compounds.

### 2.2. Physical and Chemical Indicators Detection

Water chemistry parameters were measured within 24 h after sampling. The analytical method followed the national surface water environmental quality standards established by the Chinese government (GB3838-2002). The method used for Polycyclic aromatic hydrocarbon (PAH) analysis was from “National Environmental Protection Standards of the People’s Republic of China” (HJ478-2009). The high-performance liquid chromatography (HPLC) analysis steps were as follows: the mobile phase was (A) methanol and (B) water with a ratio of 80:20 at a flow rate of 1.0 mL/min, increments of 1.2% methanol/min to 95% methanol + 5% water, hold until the peak was completed and the ultraviolet detection wavelengths were set to 254 nm, 220 nm and 295 nm.

### 2.3. Bioassays

#### 2.3.1. Acetylcholinesterase Inhibition Assay

The wastewater samples were concentrated or diluted, six concentrations (0.1×, 0.2×, 0.5×, 1×, 2× and 5×) of wastewater samples were used to generate the dose–response curve, and the concentration for 50% of maximal effect (EC_50_) was calculated via linear regression with Prism 6.0 (GraphPad Software, San Diego, CA, USA).

Methomyl was used as the positive control; the corresponding *EC*_50_ was obtained. Briefly, water samples with different dilution ratios were sequentially added to 96-well plates in triplicate, each 100 μL, and three wells were set to add an equal amount of phosphate buffer as a blank control. Then, 5, 5′-dithiobis-(2-nitrobenzoic acid), 2-(acetylthio)-N, N, N-trimethylethylammonium iodide and electric eel AChE solution were added to each dilution followed by thorough mixing on a plate shaker. After 15 min, the enzymatic reaction was measured at an OD_412_ nm in a plate reader (iD3, Molecular Devices, San Jose, CA, USA). The inhibition of AChE by the water sample was calculated according to the following formula:$$E=1-\frac{\Delta A_{t}}{\Delta A_{c}}$$
where Δ*A_t_* is the change in absorbance of the experimental group compared with the initial and Δ*A_c_* is the change in absorbance of the blank group compared with the initial one.

Toxicity equivalence factor (*TEF*) refers to the index for evaluating the relative toxicity strength or health impact degree of a compound isomer. The toxicity of environmental water samples can be evaluated more intuitively with *TEF*. The calculation method was based on the following formula:$$TEF=\frac{{EC}_{50,positive}}{{EC}_{50,sample}}$$
where *EC*_50*,positive*_ and *EC*_50*,sample*_ are half the maximum effective concentration of positive control group and sample group and sample groups.

#### 2.3.2. Cytotoxicity Assay

The wastewater samples were concentrated or diluted, and six concentrations (0.25×, 0.5×, 1×, 2×, 5× and 10×) of the wastewater from the three WWTPs were selected for testing. The human hepatocellular carcinoma cell line HepG2 was selected to detect the cytotoxicity of influents and effluents of the chemical industry plants in this study. The HepG2 cells obtained from KeyGen Biotech (Nanjing, China) were maintained in a Dulbecco’s modified Eagle medium (DMEM) supplemented with 10% fetal bovine serum (FBS). After being exposed for 24 h, the cell viability was detected using a cell counting kit-8 (CCK8) kit (Dojindo Molecular Technologies, Kumamoto city, Japan). The kit determines the number of living cells by measuring the enzymatic reaction at an OD450 nm in a plate reader (iD3, Molecular Devices). Cell viability was calculated according to the following formula:Cell viability (%) = *A*_experiment_/*A*_blank_ × 100%

Graphpad Prism software (GraphPad Software, San Diego, CA, USA) was used to fit the dose–effect relationship to find the *EC*_50_.

#### 2.3.3. Estrogenic Effect Assay

The human breast cancer cell line T47D-Kbluc (American Type Culture Collection, Rockville, MD, USA) was chosen as the indicator of estrogenic effects in wastewater samples. Roswell park memorial institute 1640 (RPMI 1640) medium was used for cell culture [11]. Activated carbon was added to reduce the estrogen concentration in the culture medium. The fetal bovine serum in the culture medium was replaced with activated carbon to adsorb the fetal bovine serum to reduce the estrogen residue in the culture medium. The cells were exposed to the cell culture fluid with water samples for 24 h and firefly luciferase was added after lysis. Luminescence was measured using a microplate reader (Synergy H1, BioTek, Santa Clara, CA, USA).

The dose–effect curve was generated according to the measured water luminescence and was used to calculate the corresponding *EC*_50_ using GraphPad Prism. β-estradiol (E2) was used as a positive control.

#### 2.3.4. Genotoxicity Assays

In this study, the Comet assay was used to detect the genotoxicity of wastewater samples. The Comet assay or single-cell gel electrophoresis assay can detect DNA damage and repair at the single-cell level through qualitative and quantitative measurement of single-strand DNA breaks, and accurately reflects the level of DNA damage and repairability. In this study, the Comet assay was used to detect the genotoxicity of wastewater samples. Algae *Euglena gracilis* was provided by the Freshwater Algae Seed Bank (FACHB) of the Typical Culture Collection Committee of the Institute of Hydrobiology, Chinese Academy of Sciences. *E. gracilis* was cultured on Checcucci medium at 25 °C with a 12 h:12 h light/dark cycle in an incubator.

A total of 1.5 mL of *E. gracilis* culture medium was centrifuged at 4000 rpm for 5 min and the precipitate was exposed to the wastewater for 30 min and centrifuged again to collect the precipitated cells. Cells were embedded in 100 μL of 1% low-melting agarose (LMA) sandwiched between 0.7% normal-melting agarose (NMA) and 1% LMA on microscope slides. The slides were placed in a lysis solution (300 mM NaOH; 30 mM Na_2_-EDTA·2H_2_O; 0.01% SDS; 9% DMSO; and 1% Triton X-100) for 20 min at 4 °C. Then, the slides were placed in a horizontal electrophoresis unit with fresh alkaline electrophoresis buffer (300 mM NaOH and 1 mM N_a2_-EDTA·2H_2_O, pH 13.0) added and the liquid level was controlled to be 2 mm above the slide at 4 °C for 20 min to allow for DNA unwinding. Electrophoresis was carried out using the same buffer for 20 min using 20 V (0.8 V/cm) and 300 mA at 4 °C. The slides were neutralized by immersing in 0.4 M Tris buffer at pH 7.5 for 5 min and stained with ethidium bromide. The slides were analyzed under a fluorescence microscope (Nikon Eclipse 50i, Tokyo Metropolitan, Japan). Comet Assay Software Project (CASP, University of Wroclaw, Wroclaw, Poland) image analysis software was used to analyze DNA damage. 

Umu/SOS experiments used the method of existing studies [12] and *Salmonella typhimurium* (TA1535/pSK1002, *S. typhimurium*) were used to detect the genotoxicity of wastewater influents and effluents. The specific experimental steps were as follows: (a) Shake the bacteria with TGA medium (tryptone 10 g/L, NaCl 5 g/L, HEPES 11.9 g/L, glucose 2 g/L; final pH adjusted to 7.0 ± 0.2) overnight for 12–16 h, dilute the bacterial solution with fresh medium 10 times the next day and continue to culture for about 1.5 h to OD_600_ = 0.2. (b) Add the diluted samples to 96-well plate A at 180 μL per well, then add 20 μL 10× medium and 70 μL bacterial solution. Add another three wells with 153 μL water, 27 μL 4-nitroquinoline-1-oxide (4-NQO), 20 μL TGA medium and 70 μL bacterial solution as positive controls. Add three wells with 180 μL water, 20 μL TGA medium and 70 μL bacterial solution as negative controls, and 180 μL water, 20 μL 10× medium and 70 μL bacterial solution as blank controls. Incubate at 37 °C for 2 h. (c) Take new plate B, add 270 μL TGA medium, preheat at 37 °C. Take 30 μL of each well in plate A and add it to the corresponding well in plate B. Incubate for 2 h and measure the absorbance of plate B at OD_600_. (d) Take new plate C, add 120 μL B buffer to each well and preheat at 28 °C. Take 30 μL of each well in plate B and add it to the corresponding well in plate C. Quickly add 30 μL of 2-Nitrophenyl β-D-galactopyranoside (ONPG), mix well and put into the incubator. After shaking at 28 °C for 30 min, add 120 μL of blocking solution to each well of plate C to stop the reaction. Measure the absorbance of plate C at OD_420_. Calculate bacterial growth factor (*G*) and induction ratio (*IR*) according to the following formula [13,14]:$$\begin{array}{l} G=\frac{\left(A_{600,S}-A_{600,B}\right)}{\left(A_{600,N}-A_{600,B}\right)}(When\;G>0.5,it\;can\;be\;used\;for\;IR\;value\;calculation)\\ IR=\frac{\left(A_{420,S}-A_{420,B}\right)}{\left(A_{420,N}-A_{420,B}\right)}\times \frac{1}{G}(When\;IR>2,the\;test\;result\;is\;judged\;to\;be\;positive)\\ \beta \;\mathrm{Galactase}\;\mathrm{activity}\left(UI\right)=\frac{\left(A_{420,S}-A_{420,B}\right)}{\left(A_{600,S}-A_{600,B}\right)} \end{array}$$

In the formula, *A*_600,*S*_ is the absorbance of water sample at 600 nm, *A*_600,*B*_ is the absorbance of blank at 600 nm and *A*_600,*N*_ is the absorbance of negative control at 600 nm; *A*_420,*S*_ is the absorbance of the water sample at 420 nm, *A*_420,*B*_ is the absorbance of the blank at 420 nm and *A*_420,*N*_ is the absorbance of the negative control at 420 nm.

Comparing the measured slope of the P-galactosidase curve in the test sample with the slope of the P-galactosidase curve of the 4-NQO sample measured simultaneously, the 4-NQO in the test sample can be obtained in an equivalent concentration (TEQ_4-NQO_):$${\mathrm{TEQ}}_{4\text{-}\mathrm{NQO}}\left(\mu g/L\right)=\frac{K_{sample}\left(\mathrm{unit}/L\right)}{K_{4\text{-}\mathrm{NQO}}\left(\mathrm{unit}/\mu g\right)}$$

### 2.4. Data Analysis

#### 2.4.1. Statistical Analysis

All the data were expressed as the mean ± standard deviation. The data analyses were performed using SPSS 13.0 and the graphics were generated and produced using Microsoft Excel and GraphPad Prism 8.0 (San Diego, CA, USA). One-way analysis of variance (one-way ANOVA) with Tukey’s post hoc test was used to assess the comparisons between two groups and the correlation between variables. For all analyses, *p* < 0.05 was considered significant, and *p* < 0.01 was considered highly significant. Data analysis was repeated three times to reduce errors.

#### 2.4.2. Health Risk Assessment

The detection adopted 16 PAH congeners listed in the “National Environmental Protection Standard of the People’s Republic of China” (HJ478-2009) as evaluation indicators. According to the American National Academy of Sciences (NAS) health risk assessment model, PAHs in industrial wastewater are harmful to human health through oral and skin exposure routes (NAS, 2004). The US Environmental Protection Agency (EPA) has classified 16 PAHs as priority pollutants based on their possible human exposure and toxicity. Most types of PAHs can diffuse through cellular membranes, resulting in toxic effects to organisms [15]. The toxicity parameters of 11 PAHs are listed in Appendix A. According to the toxicity of the target PAHs, the human health risk assessment was based on the above exposure routes (Appendix A).

## 3. Results and Discussion

### 3.1. PAHs in Wastewater Samples

A total of ten different PAHs were detected in the six samples (Table 1), and their monomer concentrations were between not detectable (ND) and 10.58 mg/L. Concentrations of NAP in influents of plants W, J and T were relatively high, reaching 10.58 mg/L, 12.07 mg/L and 9.87 mg/L, respectively. Organic substances such as fluoranthene were not detected in the influents. Still, there were trace concentrations present in the effluents, indicating that in the process of biochemical treatment, both the decomposition of organic substances and the generation of pollutants occurred. This may be a by-product of the WWT process of decomposing organic substances through photochemical and biological transformation. Correspondingly, the basic physical and chemical indicators of the chemical park wastewater along the Yangtze River changed in effluent wastewater (Appendix A). 

### 3.2. Toxicity Effects

#### 3.2.1. Neurotoxicity of Wastewater

Wastewater samples from the three wastewater treatment plants showed different inhibitory effects on AChE (Figure 2). Among them, the influent wastewater of the J plant showed the strongest inhibition, reaching 55%. The inhibition of AChE in plants W and T reached 46% and 33%, respectively. Compared with the influent samples, the inhibition rates of AChE in the effluent samples from the three water plants all decreased, indicating that the WWT process reduced the concentrations of substances with AChE inhibition activity in the samples. The inhibition rates of AChE are closely related to neuronal function disorder and death [16,17,18]. Examples of AChE inhibitors in wastewater include some low-level pollutants, such as heavy metals or detergents that are widely present in urban rivers, and the AChE inhibitors present in wastewater can adversely affect humans and animals. The toxic equivalents of the water samples were used to evaluate the neurotoxicity of wastewater samples more accurately. The *TEF* indicator can be used to analyze the environmental level of pollutants and their potential impact [19,20]. The *EC*_50_ significantly increased after the WWT process, while the *TEF* decreased (Table 2), indicating reduced neurotoxicity of the wastewater samples.

#### 3.2.2. Cytotoxicity of Wastewater 

The influents from the WWTPs showed varying concentration-dependent effects on HepG2 cells’ viability (Figure 3A). Compared with the J plant, wastewater from the W and T plants showed relatively strong cytotoxicity. According to the Industrial Wastewater Biological Toxicity Classification Standard, all influent samples from the three WWTPs are classified as cytotoxic. 

The cytotoxicity results of the effluents are shown in Figure 3B. The results showed that the WWT decreased the cytotoxicity of the wastewater. The lowered cytotoxicity in the effluent samples may be due to the partial oxidation of organic fractions by the biochemical treatment process [21,22], and the specific causative agent in the wastewater of the cytotoxicity was partially removed [23].

#### 3.2.3. Estrogenic Effect of Wastewater

Environmental endocrine disruptors are exogenous chemicals that can cause abnormalities and disorders in the endocrine system [24,25,26]. The influent group of wastewater samples showed significant estrogenic effects, and their equivalents reached 0.65 ng/L, 0.74 ng/L and 0.49 ng/L (Figure 4A), respectively. After WWT, the estrogenic effect in plants W, J and T decreased in effluent wastewater by 68%, 59% and 60%, indicating the efficiency of the treatment process. The average removal efficiency of the endocrine-disrupting effect through the WWT was 58–84%. The bacterial action of the wastewater treatment likely facilitates the degradation of endocrine disruptors [27]. Directive 2013/39/EU of the European Parliament and Council proposed a monitoring level for 17β-estradiol (E2) of 0.4 ng/L in the environment [28]. E2 is a compound hormone naturally synthesized in vertebrates which plays an important role in the endocrine and reproductive systems [29]. After the biochemical treatment, the toxic equivalents of each WWTP’s effluent were lower than this limit, indicating a low ecological risk to aquatic ecosystems (Table 3).

#### 3.2.4. Genotoxicity of Wastewater

It is well-known that the release of genotoxic substances in the environment can damage germinative cells and reduce the abundance and fertility of species in ecosystems [30,31]. Tail length (TL), Tail DNA% (tDNA%), olive tail moment (OTM) and tail moment (TM) were the main parameters of the Comet assay [32]. The four Comet assay parameters showed consistency between the influents and effluents of each sewage treatment plant (Figure 5), and the genotoxicity of the effluent was significantly reduced compared with that of the influents after the WWT. Among the measured parameters, OTM simultaneously reflects the DNA content in the Comet tail and the shape of the Comet tail, and is a commonly used indicator to quantify the degree of DNA damage [19,33]. It can be seen from Figure 5 that the OTMs of the six wastewater samples were significantly higher than those of the control, indicating that each sample caused significant DNA damage to the *E. gracilis* [34,35,36]. Similarly, the study found, using the Comet assay, that organic extracts from Taihu Lake can induce DNA damage on microalgae cells [37,38,39]. The genotoxicity of the wastewater samples was J > T > W, and the effluents showed decreases in genotoxicity by 41.06%, 36.12% and 37.15%, respectively.

Umu/SOS results showed genotoxicity of wastewater influents, and no increase in genotoxicity was observed in wastewater effluents (Appendix A). However, both influents and effluents caused growth inhibition and cytotoxicity of *S. typhimurium* (Appendix A). The results indicated that the WWT effluents still had potential genotoxicity to aquatic organisms.

### 3.3. Risk Assessment of PAHs in Wastewater Samples

The PAHs in the ambient air released by the wastewater not only cause strong odor, but also cause a threat to the health of people exposed to these substances [40,41]. According to the risk characterization model, the health risks caused by skin contact and respiratory exposure to PAHs can be calculated. The risk of PAHs from the wastewater samples in the industrial parks along the Yangtze River to human health through the respiratory route was higher than that through skin contact (Appendix A). According to the findings of the USEPA for the non-carcinogenic risk, PAHs are harmful to human health when the risk index is greater than one. Using the health risk assessment model, the calculated PAH risk index of industrial wastewater along the Yangtze River was below one, indicating that the concentrations of 11 measured PAHs in wastewater were less harmful to human health through skin contact or respiratory exposure. 

### 3.4. Correlation Analysis of Toxicity at Different Endpoints

The toxicity of complex pollutant mixtures in water can be reliably assessed only by applying a suite of bioassays. In this study, five toxicity endpoints involving the use of the HepG2 cell line, T47D-Kbluc cell, *E. gracilis*, *S. typhimurium* and electric eel AChE activity were applied. A comprehensive assessment of the exposure toxicity of the influents and effluents in the chemical parks along the Yangtze River was carried out through heatmaps (Figure 6) using cytotoxicity, AChE inhibition rate, estrogenic effect, mutagenicity, and DNA damage assessments of the influents and effluents. It is important to use different organisms or their cells, as they can produce different reactions to pollutants present in the wastewater [42]. 

Variations in the results may be caused by organism-specific responses to pollutants in the wastewater, which emphasizes the need to perform a battery of bioassays using different test organisms to perform a comprehensive toxicity and environmental risk assessment of pollutants in wastewater. Traditional physical and chemical monitoring of pollutants is not adequate to provide complete information on the potential toxic effects of pollutants on living organisms, including humans [43]. To assess genotoxicity, this study used *S. typhimurium* and *E. gracilis* as the test organisms to conduct Umu/SOS and Comet assays, respectively. The results indicated that the WWT process reduced the inhibition of AChE, estrogenic effects, mutagenicity and DNA damage. The Comet assay applied to *E. gracilis* was applicable for genotoxicity testing of industrial wastewater. *E. gracilis* can respond rapidly to various pollutants and be a bio-indicator for deteriorating water quality conditions [44]. 

The specific pollutants in wastewater have the potential to have a range of toxic effects on environmental health. There is a critical need to explore in-depth the potential ecological impacts of specific substances in wastewater and the improvement of wastewater technology for more efficient removal of substances with cytotoxic and genotoxic properties. 

## 4. Conclusions

In this study, risk assessment of the wastewater from WWTPs in chemical parks along the Yangtze River was carried out by detecting specific chemical pollutants and using a battery of bioassays to detect their toxicity. A variety of PAHs were detected in wastewater samples where relatively high concentrations of NAP, ANT and BaP were detected. The bioassays used in this study showed that the WWT process of the W, J and T plants can effectively reduce the cytotoxicity, neurotoxicity and estrogenic and genotoxic effects of the industrial wastewater. Nevertheless, all effluent wastewater from the WWTPs has been characterized by the bioassays as having potentially high ecotoxicity, indicating that their discharge into the environmental water body would potentially cause harm to aquatic organisms. This study provides theoretical support and scientific basis for the environmental risk assessment of industrial wastewater and the progress of wastewater treatment technology. It is envisaged as a guide for the application and development of future industrial wastewater risk assessment standards.

## Figures and Tables

**Figure 1 toxics-11-00702-f001:**
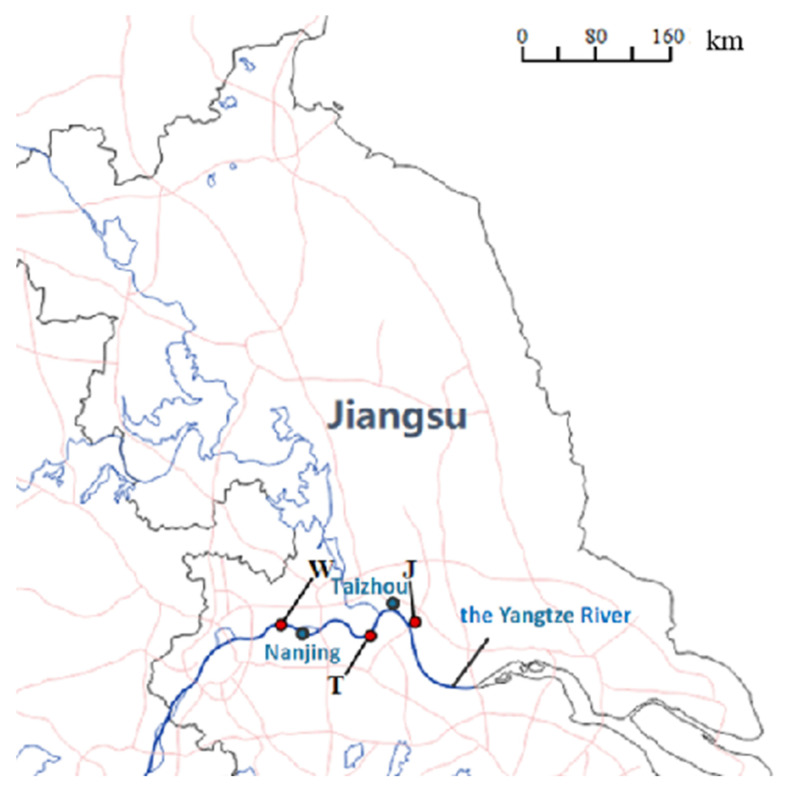
Sampling location of three WWTPs studied in this study.

**Figure 2 toxics-11-00702-f002:**
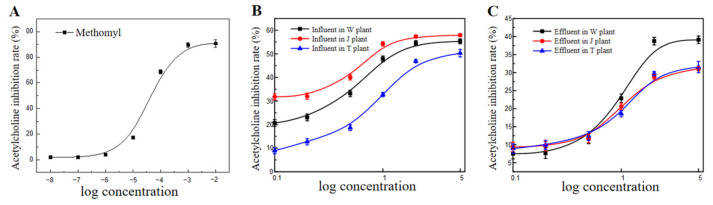
Inhibition rates of AChE in influents and effluents from three wastewater treatment plants: (**A**) methomyl, (**B**) influents, (**C**) effluents.

**Figure 3 toxics-11-00702-f003:**
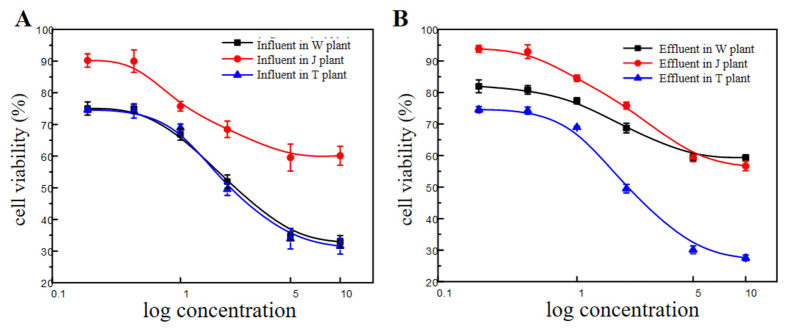
HepG2 cell viability in influents and effluents from three wastewater treatment plants: (**A**) influents, (**B**) effluents.

**Figure 4 toxics-11-00702-f004:**
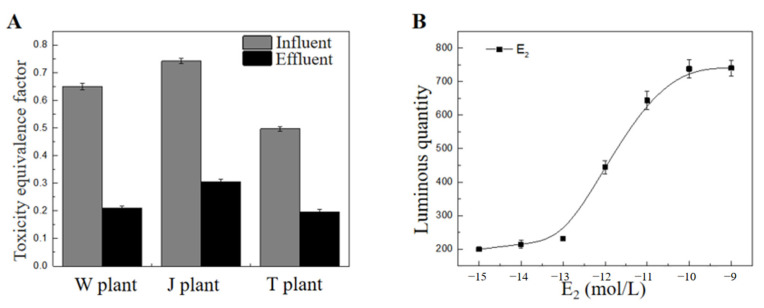
Estrogenic effect in influents and effluents from three wastewater treatment plants: (**A**) toxicity equivalence factor of influents and effluents, (**B**) 17β-estradiol (E2) dose–effect relationship.

**Figure 5 toxics-11-00702-f005:**
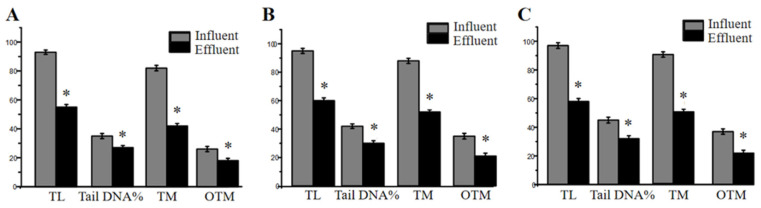
Algae *Euglena gracilis’* DNA damage in three WWTPs: (**A**) influent and effluent in W plant, (**B**) influent and effluent in T plant, (**C**) influent and effluent in J plant. * *p* < 0.05, compared with the influents. TL: Tail length. TM: Tail moment. OTM: Olive tail moment.

**Figure 6 toxics-11-00702-f006:**
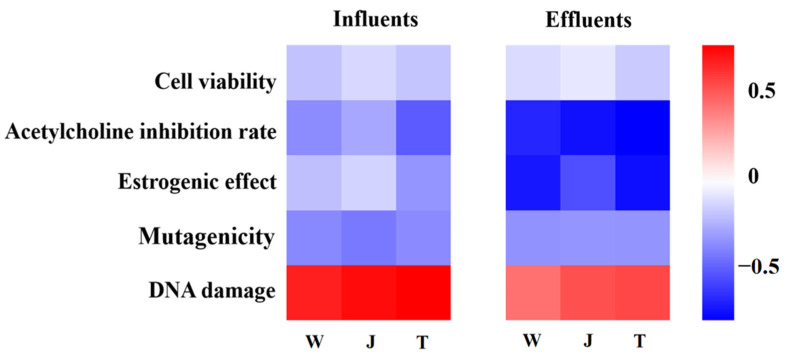
Correlation analysis of toxicity data on different test organisms for influents and effluents from three wastewater treatment plants.

**Table 1 toxics-11-00702-t001:** Concentrations of PAHs in wastewater samples.

(mg/L)	W Plant		J Plant		T Plant	
PAH	Influent	Effluent	Influent	Effluent	Influent	Effluent
Naphthalene (NAP)	10.58	0.04	12.07	0.01	9.87	0.02
Acenaphthylene (ACY)	0.06	N.D.	0.14	N.D.	0.07	N.D.
Fluorene (FLU)	0.02	N.D.	0.08	N.D.	0.05	N.D.
Fluoranthene	N.D.	0.11	0.08	N.D.	0.08	N.D.
Phenanthrene (PHE)	0.05	0.01	0.08	N.D.	N.D.	N.D.
Anthracene (ANT)	1.23	0.02	1.42	N.D.	N.D.	0.02
Benzoanthracene (BaA)	0.08	N.D.	0.19	0.01	0.04	N.D.
Benzofluoranthrene	0.06	0.01	0.25	0.02	0.03	N.D.
Benzo(a)pyrene (BaP)	1.02	0.02	1.26	0.04	1.17	0.06
Benzo(g,h,i)perylene (B(g,h,i) P)	0.05	N.D.	0.08	N.D.	0.06	N.D.
Total	13.28	0.23	15.99	0.1	11.64	0.11

Note: N.D.: Below detection limit.

**Table 2 toxics-11-00702-t002:** *EC*_50_ and *TEF* of influents and effluents from three wastewater treatment plants.

		*EC* _50_	*TEF*
W plant	Influent	3.06	2.43
	Effluent	80.54	0.10
J plant	Influent	4.04	1.84
	Effluent	65.89	0.11
T plant	Influent	6.88	1.08
	Effluent	85.35	0.10

*TEF*: Toxicity equivalence factor.

**Table 3 toxics-11-00702-t003:** Estrogenic effect *TEF* of effluent in chemical parks along the Yangtze River.

	W Plant	J Plant	T Plant	E2 Limit
Toxic equivalent factor	0.21	0.31	0.20	0.40

## Data Availability

Not applicable.

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
