# Peer review of "Using a Battery of Bioassays to Assess the Toxicity of Wastewater Treatment Plant Effluents in Industrial Parks"

_toxics, 2023, doi:10.3390/toxics11080702_

Round 1

Reviewer 1 Report

Dear authors,

The manuscript is an interesting study in the context of sustainable development, as it is necessary to ensure water for the entire population and to ensure good environmental quality. Worldwide, water pollution is one of the major environmental problems and the efficiency of water pollution control is mandatory.

I have some minor comments:

Line 22: replace was tewater with wastewater.

Line 96: please rephrase in order to avoid using the expression "followed following"….

Line 136: what represents the FBS? Please explain this abbreviation 

Line 166 you may replace precipitation with precipitate.

Line 196 What represents ONPG, please explain.

You may add some reference for bacterial growth factor G and induction ratio IR (lines 200-202) 

Line 210 what represents  K ?

Line 228 replace trget with target. 

You can add the abbreviations used in table 1 (i.e. NAP), otherwise, it is difficult to search in the supplementary material

Please make the figures bigger because it is difficult to read. 

Thus, my decision is a MINOR REVISION.

Line 22: replace was tewater with wastewater.

Line 96: please rephrase in order to avoid using the expression "followed following"….

Line 166 you may replace precipitation with precipitate.

Line 228 replace trget with target. 

Reviewer 2 Report

The manuscript entitled “Using battery of bioassays to assess the toxicity of wastewater treatment plant effluents in industrial parks” is important for several studies have investigated the aims to combine chemical and biological analysis of industrial wastewater before and after WWT, along the Yangtze River in Jiangsu Province. The basic physical and chemical properties of the wastewater samples were measured as a basis for ecological and health risk assessment of wastewater pollution in these industrialized areas. Biological assays were performed at different biological organizations to evaluate the environmental risk of wastewater from chemical industrial parks. Genotoxicity assays were performed using two organisms, freshwater algal species Euglena gracilis and Salmonella typhimurium.

The introduction is relevant but must include new references. The discussion, in the light of results and knowledge, is relevant.  

Based on these comments, I recommend a moderate revision of analytical aspects of this manuscript before final decision about its acceptance.

Minor comment: 

1 - Introduction: 

L26-72: rewrite this part with new and important references:

In the last year (Basova Marina et al.; 2022 and Parrino V. et al.; 2018), they highlighted the use of new parameters with an important tool to highlight various pollutants: 

Basova Marina et al., (2022) - Intra-Decadal (2012–2021) Dynamics of Spatial Ichthyoplankton Distribution in Sevastopol Bay (Black Sea) Affected by Hydrometeorological Factors. ANIMALS; 12, 3317; https://doi.org/10.3390/ani12233317;

Parrino V. et al., (2018) - Comparative study of haematology of two teleost fish (Mugil cephalus and Carassius auratus) from different environments and feeding habits. The European Zoological Journal, 85:1,194-200.

2 - Materials and Methods:

2.1. Sample collection and preparation

The collected samples were processed as follows: filtration (0.7 μm glass fiber filter) was performed to remove particles, followed by acidification to pH 2–3 with concentrated hydrochloric acid during the analyses, the light period was observed: dark photoperiod? Is there any previous research in these areas??...if not, clarify this aspect better!

Please describe the absence or presence of other pollutants such as detergent discharges…

Please specify the guidelines followed to carry out the validation.

Statistical:

The statistical analysis used is appropriate.

3 - Results and Discussion:

The authors should be reduce this part, please the results shown in figures.

This part is very long, reduce it and discuss only the obtained results.

4 - Conclusions

It is OK

Figures and Tables:

Rewrite in all legend (insert the species name appropriately)

Review the manuscript with a request for a minor change of the English language
